# The Association of *IFNL4* Gene Polymorphisms with Hepatitis B Virus (HBV) Infection in the Northern Region of Pará, Brazil

**DOI:** 10.3390/ijms251910836

**Published:** 2024-10-09

**Authors:** Álesson Adam Fonseca Andrade, Carolina Cabral Angelim, Letícia Dias Martins, Amanda Roberta Vieira Sacramento, Renata Santos de Sousa, Raissa Lima Correa, Simone Regina Souza da Silva Conde, Antonio Carlos Rosário Vallinoto, Rosimar Neris Martins Feitosa, Greice de Lemos Cardoso Costa

**Affiliations:** 1Human and Medical Genetics Laboratory, Institute of Biological Sciences, Federal University of Pará, Belém 66075-110, Brazil; prof.alessonandrade.bio@gmail.com (Á.A.F.A.); carolina.angelim@icb.ufpa.br (C.C.A.); lemartias@hotmail.com (L.D.M.); roberta.amanda@hotmail.com (A.R.V.S.); raissa.correa@icb.ufpa.br (R.L.C.); 2Postgraduate Program in Biology of Infectious and Parasitic Agents, Federal University of Pará, Belém 60000-000, Brazil; renatadeeesousa@gmail.com (R.S.d.S.); vallinoto@ufpa.br (A.C.R.V.); rosimar@ufpa.br (R.N.M.F.); 3Postgraduate Program in Virology, Evandro Chagas Institute, Ananindeua 67030-000, Brazil; 4Laboratory of Virology, Institute of Biological Sciences, Federal University of Pará, Belém 66075-110, Brazil; 5Faculty of Medicine, Institute of Health Sciences, Federal University of Pará, Belém 66055-080, Brazil; sconde@ufpa.br

**Keywords:** HBV, *IFNL4*, polymorphism, SNPs, predisposition

## Abstract

It is heavily suggested that one *IFNL4* gene polymorphism, rs12979860 (T/C), exerts influence on the outcome of HBV infection, with the rs12979860-T allele being classified as a risk predictor, and the rs12979860-C allele being classified as a protective one. This study investigated whether the rs12979860 *IFNL4* gene polymorphism presented any association with the clinical severity for HBV carriers in an admixed population in Northern Brazil. A total of 69 samples were investigated from infected people from the city of Belém-Pará. The rs12979860-T allele was positively associated with HBV infection, suggesting a higher risk of chronicity. This research’s importance is that the polymorphism influence was investigated in a population of HBV carriers with a heterogeneous genetic profile, formed through the extensive admixture of different ethnic groups, including Europeans, Africans, and Natives with indigenous heritage. This analysis is particularly important since highly mixed populations do not always follow the same association patterns previously established by studies using populations classified as more genetically homogeneous, due to a different formation process.

## 1. Introduction

Globally, approximately 296 million people are chronically infected by hepatitis B virus (HBV), with around 1.5 million new infections and 820,000 deaths annually, due to the burden of this infection [1]. In Brazil, from 2000 to 2022, 276,646 cases of hepatitis B were confirmed, with the chronic phase being predominant until 2020 (73.1%) [2].

The immune system response to HBV infection happens through the recognition of pathogen-associated molecular patterns (PAMPs)—which, via various intracellular signaling pathways, activate pro-inflammatory and antimicrobial responses. Effector elements, such as various interferons (IFNs), degrade viral components, for example, DNA, RNA, and proteins [3,4].

Polymorphisms in the *IFNL4* gene may be determinants in modulating the immune response and, consequently, the outcome of infectious diseases [5]. Although some studies have not shown a statistically significant correlation between the SNP rs12979860 (T/C) and HBV infection [6,7,8,9,10,11,12,13], there is also evidence that the presence of rs12979860 influences the immune response against HBV at different stages of the infection [14,15,16,17].

Most of the previous research investigating these genetic variants and their associations with HBV infection has been conducted in genetically homogeneous populations, where there has not been an admixture of different population groups during their formation. However, this study investigated an admixed population from Northern Brazil, formed through the contributions of Native People of indigenous heritage, Europeans, and Africans, which could exert influence on modifying the illness’s behavior profile from a genetic variability standpoint, caused by this specific process of populational formation.

This research investigated the frequency of the IFNL4 rs12979860 gene polymorphism, associating it with the clinical severity of the HBV infection, in a mixed population from the North Region of Brazil.

## 2. Results

From a total of 69 HBV carriers, 61% (*n* = 42) were male, 39% (*n* = 27) were female, and the average age was 48 years old (Appendix A: Sociodemographic characterization of people with HBV). Most of these HBV carriers were classified as asymptomatic (51%), with 61% (*n* = 42) presenting no clinical signs of cirrhosis, and 62% (*n* = 43) being at the infection’s chronic stage. The viral load was still detectable in 65% of the total individuals. Most of them had normal levels of AST/TGO (64%), ALT/TGP (70%), and GGT (59%).

The clinical and laboratory profiles, categorized according to the included patient subgroups, are described in Table 1.

After analyzing the allelic and genotypic frequencies, it was found that most of the individuals had heterozygous rs12979860-TC (*n* = 44; 63.8%), with the rs12979860-T presenting a higher frequency. All the frequencies stated were further confirmed using the Hardy–Weinberg Test, and they are detailed in Table 2.

For comparison purposes between the population included in this study and a healthy one with no HBV infection, the data used came from the literature. The data of this study’s population were compared to the data previously published by Amaral [18], with this cited publication presenting results linking the polymorphism to a healthy population with no HBV infection. When we analyzed the genotypic frequencies either in isolation or simultaneously, in both groups, we were able to find statistically significant results (*p* value = 0.0114) (Table 3).

These genotypic frequencies were compared according to the dominant model (*p*= 0.0037) and the recessive model (*p* = 0.3879) between the samples. The dominant model shows that the homozygous genotype for the rs12979860-C allele had the lowest frequencies in the HBV carrier sample (Table 3).

After this step, we also verified whether there was any possible association between the rs12979860 and the progression/severity of HBV in the individuals comprising our study sample. The results show no statistical significance (*p* value = 0.9873) (Appendix A: Paired comparison (Fisher’s exact test) of different genotypes and clinical manifestations).

Table 4 shows the results generated when this study associated chronic HBV infection and the control group samples [18]. First, this specific analysis occurred in isolation and did not show statistical significance (*p* value = 0.1592) (Appendix A. Paired comparison between chronic HBV vs. control groups (Chi-square test)). After observing that the rs12979860-T allelic frequency remained high throughout our sample of HBV infection carriers, a comparison between genotypes—the combined genotypes with the T allele (namely, TT and TC) versus the one that is homozygous rs12979860-C—was also carried out, the results of which are statically significant. This confirms that the combination of genotypes with the rs12979860-T allele is indeed prevalent within the group of chronically infected carriers (*p* value = 0.0289).

The sample of HBV carriers, when divided into two groups, according to the presence of cirrhosis, was not trustworthy from a statistical analysis standpoint due to the small number of samples from individuals with cirrhosis (*n* = 7). The results of the comparison between the genotypic frequencies and these results are not significant (*p* value = 0.5742).

The laboratory aspects, such as the carriers’ HBV viral load, were also analyzed in relation to the *IFNL4* variants present in the study population. The median value for the viral load of the 69 total samples was 104 UI/mL (classified as not elevated). After conducting a Mann–Whitney test, we were unable to observe any statistically significant association (*p* value = 0.7646) (Appendix A: A paired comparison of the different *IFNL4* genotypes and the levels of HBV DNA, AST/TGO, ALT/TGP, and GGT (Mann–Whitney)).

The main biochemical markers for liver damage were also investigated—AST/TGO, ALT/TGP, and GGT—in relation to the genotypes (Appendix A: A paired comparison of the different *IFNL4* genotypes and the levels of HBV DNA, AST/TGO, ALT/TGP, and GGT (Mann–Whitney)). The results of the Mann–Whitney comparison show no association between the different *IFNL4* genotyping profiles and the levels of AST/TGO (*p* value = 0.3271), ALT/TGP (*p* value = 0.3271), or GGT (*p* value = 0.1781).

## 3. Discussion

The majority (62%) of HBV-infected individuals in the present study were in the chronic phase, and although most infections by this virus are usually asymptomatic, in Brazil, there is a higher prevalence of chronic hepatitis B [19,20].

Dai and collaborators [21] analyzed the biochemical profile and viral load of HBV carriers with chronic infection (dividing them into groups with and without hepatic steatosis) and argued that a decrease in HBV viral load levels and ALT/SGPT, as well as an increase in AST/SGOT and GGT levels, are associated with the development of hepatic fibrosis and advanced hepatic fibrosis. The population in the present study showed detectable levels of HBV DNA and normal values for the three analyzed biochemical markers.

Low genotypic frequencies of the rs12979860-C homozygote allele were observed in the sample of HBV carriers in this study. This result suggests that the high frequencies of the homozygous and heterozygous genotypes for the rs12979860-T allele in the analysis samples may suggest the association of this allele with viral infection, as discussed in previously published studies [14,15,16,17].

There was no association between rs12979860 and the variables of (i) clinical symptoms, (ii) cirrhotic status, (iii) serum levels of HBV DNA, and (iv) liver damage markers such as ALT/SGPT, AST/SGOT, and GGT. These results differ from those of other studies [22,23], likely because the genotypic frequencies in the sample of HBV-infected individuals described here in the North Region of Brazil are similar to those observed among Africans [24], while the other studies were conducted in European and Asian populations. This suggestion is also backed up by the data from Chihab and collaborators [25], who, in their study, investigated HBV carriers in Morocco and obtained results like ours.

Studies on these variants associated with HBV infection remain controversial, and most of the existing research is limited to genetically homogeneous populations. It is important to acknowledge that the genetic contribution of different ethnic groups, under a populational standpoint, can act as one of the determinants for higher or lower rates of susceptibility to various diseases, particularly infectious ones [26,27,28].

Therefore, it is necessary to investigate the distribution of polymorphisms in different continental and mixed populations so that effective public health strategies can be designed according to the needs of population groups with specific disease profiles, prompting discussions about other forms of disease management and decision making with the best interest for patient care.

Future studies will hopefully be able to continue investigating the molecular aspects of this population. Since the process of obtaining samples is based on the informed consent and willingness of carriers to participate, in addition to the authorization of patient-care institutions, another study may be capable of working with a bigger sample size, thus proposing the safest analyses on the probable associations, as suggested by previous studies with smaller sample sizes.

## 4. Materials and Methods

### 4.1. The Characterization of the Study Group and the Process of Obtaining Samples

This was an observational, transversal, and retrospective study that investigated 69 samples of HBV carriers collected from the following institutions: Hospital Universitário João de Barros Barreto (HUJBB), Centro de Atenção à Saúde em Doenças Infecciosas Adquiridas (CASA DIA), and Fundação Santa Casa de Misericórdia do Pará (FSCMPA). The samples collected at FSCMPA were obtained from 2014 to 2016, and those from HUJBB and CASADIA were collected in 2023 and 2024.

This study was conducted in accordance with the Declaration of Helsinki, and approved by the Ethics Committee of the FSCMPA (CEP, CAAE: 31223214.2.3001.5171, approved on 29 June 2022), HUJBB (CEP, CAAE: 65072922.6.3001.0017, approved on 10 April 2023), and UFPA—Instituto de Ciências da Saúde da Universidade Federal Do Pará (CEP, CAAE: 65072922.6.0000.0018, approved on 7 June 2022) for studies involving humans.

After a medical evaluation, the HBV carriers were then invited to be part of this research. The sociodemographic, clinical, and laboratory information was obtained through both questionnaires and medical records, either from the agreeing patient or by using the institutional digital record system, all upon authorization from the institutions involved. From each individual participant, an 8 mL sample of whole blood was collected in a tube containing EDTA as an anticoagulation agent, which enabled the conduction of serological tests and DNA extraction, this latter being part of the molecular biology techniques.

The adopted inclusion criteria for this research were (i) being older than 18 years of age; (ii) signing the consent form (TCLE, Termo de Consentimento Livre e Esclarecido); (iii) authorizing the collection of biological material; (iv) answering the sociodemographic questionnaire; and lastly, (v) having a confirmed HBV diagnosis, observed through the positivity of HBsAg.

Individuals excluded from this research presented the following situations: (i) not adequately answering the sociodemographic questionnaire; (ii) not having a full data set for the variables analyzed (for example, a lack of laboratory information, which would limit some of our statistical testing).

The individuals included in the sample population obeyed all the following criteria: (i) a confirmed diagnosis of HBV infection; (ii) over 18 years of age; (iii) explicit signature on the informed consent form; and (iv) proper answers to the sociodemographic questionnaire. Of the study sample, male patients comprised 61% (*n* = 42), with an age average of 48 years. Additionally, 30.5% of the total participants declared themselves as brown, regarding ethnicity (*n* = 21), and regarding residency information, most of the assessed individuals lived in Belém’s Metropolitan Region (71%, *n* = 49).

All the sociodemographic data can be seen in the Appendix A (Appendix A: Sociodemographic characterization of people with HBV).

### 4.2. DNA Extraction and Purification

The HBV carriers’ DNA extraction was conducted using the Wizard Genomic DNA Purification from Promega^®^ (Madison, WI, USA) kit, which extracts and purifies DNA from a whole blood sample. The procedure followed the standard protocol described by the manufacturer, with a minimum of 20 ng/μL for each sample.

### 4.3. The Genotyping of rs12979860 of the IFNL4 Gene

For genotyping purposes after obtaining the extracted DNA, a real-time polymerase chain reaction, or RT-PCR, was conducted, according to the protocol from the manufacturers and using the Applied Biosystems 7500 equipment (South San Francisco, CA, USA). The chosen reagents were manufactured by TaqMan™ (namely the Universal PCR Master Mix), and fluorescent probes from ThermoFischer^®^ (Carlsbad, CA, USA) were also included to function as targets for the amplification of the selected polymorphism, rs12979860.

### 4.4. Statistical Analysis

The allelic and genotypic frequencies of the polymorphism within the HBV carrier population were obtained by simple counting. The Hardy–Weinberg equilibrium (HWE) and the Chi-Squared (X^2^) tests were applied to investigate if the genotypic frequencies of the polymorphism, studied in a group of HBV carriers, were in optimal equilibrium conditions.

A comparison between the sample from this study and control individuals was carried out using data from the study published by Amaral [18], where the selected individuals were also from an admixed population within Belém, PA, without HBV infection. The Chi-squared and Fisher’s exact test methods were used to assess the categorical variables (clinical manifestations, cirrhosis development, and clinical evolution/phase) of the HBV-infected study population versus the control group and to assess the comparison between the polymorphism and the HBV clinical data. Comparisons using the dominant model (TT + TC versus CC) and the recessive model (CC + TC versus TT) were also performed within the study sample group, to observe whether genotypic combinations could be influencing the results.

When discussing the laboratory characteristics related to hepatitis B (HBV DNA, AST/TGO, ALT/TGO, and GGT), the reference values considered in this research followed the guidelines previously established by the Brazilian Health Ministry [19]. To analyze these variables, the Mann–Whitney test was conducted. However, some carriers did not have the full set of clinical information (clinical manifestations, cirrhosis, illness evolution/clinical phase) and/or laboratory parameters. Therefore, in this analysis set, the individuals without information on some variables were not considered, which explains why our sample size (*n*) is different between the groups.

The aforementioned analyses were performed using the following software programs: BioEstat 5.3 and 2024 GraphPad Prism 8.0.1 software. In all statistical instances, the significance corresponded to *p* ≤ 0.05.

## 5. Conclusions

The sample of HBV carriers from the state of Pará showed high frequencies of the rs12979860-T allele, when compared to the sample of individuals without this viral infection. There was no association between rs12979860 and the clinical symptoms, cirrhotic status, serum levels of HBV DNA, and liver damage markers. Future studies should increase the sample size and test new hypotheses about host genetics and factors that may influence the clinical progression of HBV in the state of Pará.

## Figures and Tables

**Table 1 ijms-25-10836-t001:** Clinical and laboratory profile (AST/TGO, ALT/TGP, and GGT) of people carrying HBV.

*IFNL4*	rs12979860
	*n* = 69 (%)
**Manifestations**	
Symptomatic	11 (16%)
Asymptomatic	35 (51%)
Uninformed	23 (33%)
**Cirrhosis**	
With cirrhosis	7 (10%)
No cirrhosis	42 (61%)
Uninformed	20 (29%)
**Clinical phase**	
Acute	2 (3%)
Chronic	43 (62%)
Uninformed	24 (35%)
**HBV viral load**	
Elevated	11 (16%)
Not elevated	34 (49%)
Undetectable	11 (16%)
Uninformed	13 (19%)
**AST (TGO)**	
High	20 (29%)
Normal	44 (64%)
Uninformed	5 (7%)
**ALT (TGP)**	
High	15 (22%)
Normal	48 (70%)
Uninformed	6 (9%)
**Gamma GT (GGT)**	
High	10 (14%)
Normal	41 (59%)
Uninformed	18 (26%)

AST/TGO: aspartate aminotransferase/oxalacetic transaminase; ALT/TGP: alanine aminotransferase/glutamic pyruvic transaminase; Gamma GT (GGT): gammaglutamyltransferase.

**Table 2 ijms-25-10836-t002:** Allelic and genotypic frequencies of *IFNL4* variants within this study’s HBV carrier population. The *p*-value refers to the Hardy–Weinberg equilibrium test.

rs12979860	*n* = 69 (%)	*p*-Value
TT	14 (20.3%)	*p* = 0.2569
CC	11 (15.9%)	
TC	44 (63.8%)	
T *	72 (52.17%)	
C **	66 (48%)	

*: Mutant allele; **: reference allele.

**Table 3 ijms-25-10836-t003:** A paired comparison (Chi-square test and Fisher’s exact test) of the different genotypes in the HBV group vs. the control.

*IFNL4*	Allele Genotype	HBV *n* = 69 (%)	Control *n* = 85 (%)	HBV vs. Control ^1^
rs12979860	TT	14 (20%)	12 (14%)	*p* = 0.0114
	TC	44 (64%)	41 (48%)	
	CC	11 (16%)	32 (38%)	
	T *	52%	38%	
	C	48%	62%	
	**Allele** **Genotype**	**HBV** ***n* = 69 (%)**	**Control** ***n* = 85 (%)**	**HBV vs. Control ^2^**
rs12979860	TT + TC	55 (84%)	53 (62%)	*p* = 0.0037
	CC	11 (16%)	32 (38%)	
	T *	52%	38%	
	C	48%	62%	

* Minor allele frequency (MAF); ^1^ Chi-square test; ^2^ Fisher’s exact test.

**Table 4 ijms-25-10836-t004:** A paired comparison between the control group vs. chronic HBV groups. The control analysis used Fisher’s exact test.

*IFNL4*	Allele Genotype	Chronic HBV *n* = 43 (%)	Control *n* = 85 (%)	Chronic HBV vs. Control
rs12979860	TT + TC	35 (81%)	53 (62%)	*p* = 0.0289
	CC	8 (19%)	32 (38%)	
	T	51%	38%	
	C *	49%	62%	

* Minor allele frequency (MAF).

## Data Availability

The data are contained within the article and Appendix A.

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
