# Peer review of "The Association of IFNL4 Gene Polymorphisms with Hepatitis B Virus (HBV) Infection in the Northern Region of Pará, Brazil"

_ijms, 2024, doi:10.3390/ijms251910836_

Round 1
Reviewer 1 Report
Comments and Suggestions for Authors
Analysis of the of the effect different immune system related gene polymorphisms have on the progress of severity of the symptoms is of high importance for the better understanding the pathogenesis of various infectious disease. Furthermore, it helps adjust the treatment or with the development of novel therapy options. The manuscript: Association between IFNL4 gene polymorphisms and Hepatitis B Virus (HBV) infection in the State of Pará, Northern Brazil may have some interesting data to present but flaws of this paper's presentation are multiple.
1. Stud aim is not clearly stated, if the authors are investigating just the ratios of the polymorphisms then healthy controls are missing, or at least they are not distinctly stated in the text
2. Small size of the study group is the first issue here, but the inclusion/exclusion criteria are not clearly stated, nor why the second polymorphism is tested for only a half of the study group (assuming the same participants were involved, I apologize to the authors if I missed that information somewhere in the text).
3. Dividing a small study group to begin with into patients with chronic and acute infection (the other consisting of just two patients) is making any conclusion drawn from that data set invalid (tables 3. and 4. in the supplementary material)
3. In the table 1. there are a lot of data inaccurately presented and need explanation, for example, when was the viral load measured and wat amount was high, what means "not elevated", and what uniformed (I assume the authors were trying to say there are no data, however, than those individuals should be excluded from the study).
4. English language presents a serious problem for the authors, the help of the English language professor or a native speaker is essential for this manuscript's successful publication.
To conclude, I encourage the authors to resubmit this manuscript after all the issues stated above have been resolved.
Comments on the Quality of English LanguageEnglish language present a serious challenge for the authors and before resubmission an English language professor or a native speaker needs to be consulted.
Author Response
Analysis of the of the effect different immune system related gene polymorphisms have on the progress of severity of the symptoms is of high importance for the better understanding the pathogenesis of various infectious disease. Furthermore, it helps adjust the treatment or with the development of novel therapy options. The manuscript: Association between IFNL4 gene polymorphisms and Hepatitis B Virus (HBV) infection in the State of Pará, Northern Brazil may have some interesting data to present, but flaws of this paper's presentation are multiple.
- Study aim is not clearly stated, if the authors are investigating just the ratios of the polymorphisms, then healthy controls are missing, or at least they are not distinctly stated in the text
We appreciate your comments and we tried solving the aim problem, by inserting the text “However, this study investigates an admixed population from Northern Brazil, formed through the contributions of Native People of indigenous heritage, Europeans and Africans; which could exert influence on modifying the illness behavior profile, from a genetic variability standpoint, caused by this specific process of populational formation. This research investigated the frequency of the IFNL4 rs12979860 gene polymorphism, associating it with the clinical severity of the HBV infection, within a mixed population from the North of Brazil”, lines 55 - 59.
Regarding the healthy control group, this study did not use samples characterized as “healthy individuals”, because we had no access to these. As a comparison group, we used the data previously published by Amaral (2015) (Reference: AMARAL, I. do S.A. A INFLUÊNCIA DOS POLIMORFISMOS NOS GENES INTERFERONS LAMBDA 3 LAMBDA 4 E ANCESTRALIDADE GENÉTICA NA INFECÇÃO CRÔNICA PELO VÍRUS DA HEPATITE C E NA RESPOSTA AO TRATAMENTO EM UMA POPULAÇÃO MISCIGENADA DE BELÉM-PARÁ-BRASIL. Tese (Doutorado), Universidade Federal do Pará: Belém, 2015), since this paper was also published using individuals from our state (Pará). We highlighted this in the methodology section, by writing: “The comparison between the sample from this study and control individuals, was carried out by using the study published by Amaral [19], where the selected individuals were also from an admixed population within Belém - PA, without HBV infection.” lines 251 - 253, on the topic 4.4. Statistical Analysis.
We also tried to make the control group comparison clearer by inserting the text “For comparing purposes between the population included in this study versus a healthy one, with no HBV infection, the data used came from literature” lines 100 – 101, and “The results were able to show statistical significance, which could suggest that the rs12979860-T allele, under any genotypic form, could be positively associated with HBV infection, within our sample (p value=0.0037) (Table 3).”, lines 108 - 110, under the topic of Results.
- Small size of the study group is the first issue here, but the inclusion/exclusion criteria are not clearly stated, nor why the second polymorphism is tested for only a half of the study group (assuming the same participants were involved, I apologize to the authors if I missed that information somewhere in the text).
R= We agree that our sample size is indeed an issue to be addressed, therefore we stated it explicitly on the lines 189 - 195, by the end of discussion, through the text insertion of “Future studies will hopefully be able to continue investigating the molecular aspects of this population. Since the process of obtaining samples is based on the informed consent and willingness of carriers to participate, in addition to the authorization of patient-care institutions, some other study may be capable of working with a bigger sample size; therefore, proposing safest analyses on the probable associations, suggested by previous studies with smaller sample sizes”.
The inclusion/exclusion criteria are now more evident in the text “The individuals included in the sample population obeyed all the following criteria: (i) a confirmed diagnosis of HBV infection, (ii) over 18 years of age, (iii) explicit signature on the informed consent form and (iv) proper answers to the sociodemographic questionnaire.” on the lines 221 - 224, under the subtopic 4.1. Characterization of the study group and the process of obtaining samples.”
The polymorphism rs368234815 was tested for all samples, however due to its nature being that of a dinucleotide polymorphism, we struggled with clearly and safely analyzing its genotyping pattern, with an acceptable level of trust. Thus, we discussed throughout the writing process of this manuscript, about the possibility of including or excluding these results, and at first, we ended up deciding on describing everything we have regarding this sample. But, after the reviewers' comments, the best course of action is to remove these results from the manuscript.
- Dividing a small study group to begin with into patients with chronic and acute infection (the other consisting of just two patients) is making any conclusion drawn from that data set invalid (tables 3. and 4. in the supplementary material)
We agree with the reviewers’ opinion and, once again, in the manuscript discussion, we chose to show all the analyses that were performed. However, after understanding the feedback, we took it out of both the manuscript text and supplementary material.
- In the table 1. there are a lot of data inaccurately presented and need explanation, for example, when was the viral load measured and wat amount was high, what means "not elevated", and what uniformed (I assume the authors were trying to say there are no data, however, than those individuals should be excluded from the study).
The classification of all the information included in table 1, follows the reference values previously published by the guidelines of our local Health Ministry, (Reference 20: Saúde, M. DA PROTOCOLO CLÍNICO E DIRETRIZES TERAPÊUTICAS PARA HEPATITE B E COINFECÇÕES; Ministério da Saúde: Brasília, 2017). By following these guidelines, the viral load is considered: (i) not elevated: >10<19.999 UI/ml; and (ii) elevated >20.000 UI/ml. Regarding these classified as “Uninformed”, this group encompassed individuals with no available data for this specific variable. This group was only mentioned on table 1, but when conducting the research statistical analysis, their exclusion was automatic. This is explained with the sentences “The reference values considered on this research, followed the guidelines previously established by the Brazilian Health Ministry [20]. To analyze these variables, the Mann-Whitney Test was conducted. However, some carriers did not have the full set of clinical information (clinical manifestations, cirrhosis, illness evolution/clinical phase) and/or laboratory parameters. Therefore, in this analysis set, the individuals without information on variables were not considered, which explains the reason why our sample size (n) is divergent in between groups” lines 259 - 265, topic 4.4. Statistical Analysis.
- English language presents a serious problem for the authors, the help of the English language professor or a native speaker is essential for this manuscript's successful publication.
The original text was translated to English and reviewed once again (certificate attached with the manuscript).

Reviewer 2 Report
Comments and Suggestions for Authors
The study is conducted incomprehensively. The number of cases is quite small, and the data are not concrete. Certain sentences are required to clarify and some of the words are mistyped and some mistakenly punctuations.
Comments on the Quality of English LanguageCertain sentences are required to clarify and some of the words are mistyped and some mistakenly punctuations.
Author Response
The study is conducted incomprehensively. The number of cases is quite small, and the data are not concrete. Certain sentences are required to clarify and some of the words are mistyped and some mistakenly punctuations.
We appreciate your feedback. A major revision of the manuscript was performed, and we hope that now these mistakes are corrected. We agree on the issue of having a small sample size, which can be explained, due to the fact that this study was conducted relying on brief intervals of time in which sample collection was authorized — because of operational problems, mostly related to the partner institutions (the patient care reference centers). That’s why we added a small section on the manuscript: “Future studies will hopefully be able to continue investigating the molecular aspects of this population. Since the process of obtaining samples is based on the informed consent and willingness of carriers to participate, in addition to the authorization of patient-care institutions, some other study may be capable of working with a bigger sample size; therefore, proposing safest analyses on the probable associations, suggested by previous studies with smaller sample sizes”, lines 189 - 195, by the end of the discussion.
The English manuscript was reviewed through MDPI Author Services (certificate attached with the manuscript).

Reviewer 3 Report
Comments and Suggestions for Authors
This manuscript demonstrated that patients with the rs12979860-T allele of the IFNL4 gene variant in a mixed population in northern Brazil were positively associated with HBV infection and had a higher risk of developing chronic disease, whereas the rs368234815-ΔG allele was important in the development of cirrhosis in a group of HBV-infected carriers. Although different from most studies in the current literatures, this study is important for investigating the effect of IFNL4 polymorphisms on HBV infection. However, there are a number of issues that need to be addressed in this manuscript:
1. Table 1: In the comparison of HBV vs. control in Table 1, it is known that CC genotype is the common genotype of the control samples, but why we compare TT+TC with CC to get the P-value, instead of comparing TT or TC alone with CC, please answer; why you do not compare T* and C with CC?
2. For the comparison of chronic HBV vs. control group in Table 2, why compare GG/∆∆+TT/∆G with TT/∆∆ to get the P-value, instead of comparing GG/∆∆ or TT/∆G with TT/∆∆ alone. Why not compare ∆G* and TT with TT/∆∆?
The comparison of with cirrhosis vs. no cirrhosis in Table 5 of the experiments in this paper, in the conclusion ''we were able to associate the GG/∆∆ genotype and the presence of cirrhosis, at least within this evaluated and included population (p=0,0143). However, the sample numbers in Table 5 with cirrhosis is so small that I do not think it is possible to draw this conclusion, at least the sample size should be larger to support this argument.
Author Response
This manuscript demonstrated that patients with the rs12979860-T allele of the IFNL4 gene variant in a mixed population in northern Brazil were positively associated with HBV infection and had a higher risk of developing chronic disease, whereas the rs368234815-ΔG allele was important in the development of cirrhosis in a group of HBV-infected carriers. Although different from most studies in the current literatures, this study is important for investigating the effect of IFNL4 polymorphisms on HBV infection. However, there are a number of issues that need to be addressed in this manuscript:
- Table 1: In the comparison of HBV vs. control in Table 1, it is known that CC genotype is the common genotype of the control samples, but why we compare TT+TC with CC to get the P-value, instead of comparing TT or TC alone with CC, please answer; why you do not compare T* and C with CC?
We appreciate the comments you made and chose to showcase the comparison between the mutant allele (T) genotypic combinations and the reference allele (C) when homozygous. In our results, the T allele was the more frequent one in the HBV carriers' samples, and that’s why we directed the analyses towards investigating whether the presence of both genotypes (homozygous and heterozygous) could imply statistical significance or not (together or separetely). And we were able to find statistically significant data. However, if even after these revisions, the text is still unclear to you, let us know, so we can get rid of the grouped analysis and keep only the ones that assess the genotypes individually.
We also left this information explicit on the results section, making it more clear after the text modifications: “This study population was compared to the data previously published by Amaral [19], with this cited publication presenting results linking the polymorphism to a healthy population, with no HBV infection. When we analyzed the genotypic frequencies either isolated or simultaneous, on both groups, we were able to find statistically significant results (p value=0.0114) (Table 3). Since the mutant allele frequency was prevalent in the samples coming from infection carriers, we investigated the combo of genotypes with this allele (TT or TC), against the homozygote for the reference allele (CC). The results were able to show statistical significance, which could suggest that the rs12979860-T allele, under any genotypic form, could be positively associated with HBV infection, within our sample (p value=0.0037) (Table 3).,”, lines 101 - 110. Table 3 was also modified to include the two modes of genotypic analysis performed: first, the information about the isolated genotypes, and after, the analysis on combined genotypes.
At line 121 - 129, we inserted the text “At first, this specific analysis occured isolatedly and did not show statistical significance (p value=0.1592) (Table S3. Paired comparison between chronic HBV vs. Control groups (Chi-square Test). After observing that the rs12979860-T allelic frequency remained high throughout our sample of HBV infection carriers, the comparison between genotypes — the combined genotypes with the T allele (namely, TT and TC) versus the one that is homozygous rs12979860-C — was also carried out, and this one returned with results that were significant, statistics wise; which allows the confirmation that the combination of genotypes with the rs12979860-T allele presence is indeed prevalent within the group of chronically infected carriers (p value=0.0289).” Table 4 was also modified to hopefully make the analysis conduction clear.
And lastly, in the lines 133 - 136, there is the text “The sample of HBV carriers, when divided into two groups, according to the presence of cirrhosis, were not trustworthy from a statistical analysis standpoint, due to the small number of samples from individuals with cirrhosis (N=7). The comparison between the genotypic frequencies and these results was not significant (p value=0.5742).” Table 5 was removed from the text, and to replace it, we highlighted that the sample size was deemed insufficient, and not a good parameter to consider this result as relevant.
- For the comparison of chronic HBV vs. control group in Table 2, why compare GG/∆∆+TT/∆G with TT/∆∆ to get the P-value, instead of comparing GG/∆∆ or TT/∆G with TT/∆∆ alone. Why not compare ∆G* and TT with TT/∆∆?
We appreciate your feedback, and for this specific table, we followed the same logic used for rs12979860, but the sample size in this case was considerably smaller. After the journal feedback, the authors discussed whether the rs368234815 polymorphism analysis would be kept or removed, and we chose the latter. Mostly due to small sample size and in order to improve the manuscript readability.
The comparison of with cirrhosis vs. no cirrhosis in Table 5 of the experiments in this paper, in the conclusion ''we were able to associate the GG/∆∆ genotype and the presence of cirrhosis, at least within this evaluated and included population (p=0,0143). However, the sample numbers in Table 5 with cirrhosis is so small that I do not think it is possible to draw this conclusion, at least the sample size should be larger to support this argument.
After the considerations sent by the journal, the authors also discussed about inclusion or exclusion of this polymorphism, and the final consensus was to keep the text focused on rs12979860, highlighting the small sample size as a limitation of our study. Even though the n is not as big as it would be desirable, we still deem this polymorphism as relevant, given the clinical severity associated between this variant and HCV infection. However, the tables around this polymorphism were removed, as well as the ones for rs368234815.
The English manuscript was reviewed through MDPI Author Services (certificate attached with the manuscript).

Round 2
Reviewer 1 Report
Comments and Suggestions for Authors
The authors have improved the manuscript: "Association of IFNL4 gene polymorphisms with Hepatitis B Virus (HBV) infection in the Northern Region of Pará, Brazil" to some extent, however, some issues remain. The first is use of healthy control data from a different source, if the authors had no access to the healthy samples then they should have compared the HBV positive individuals carrying C versus T allele, or compare the 3 genotypes, which brings me to the second issue. Why are two genotypes grouped (TT and TC) while CC is separate? Concluding that T allele can be associated with higher susceptibility to HBV from this data set is somewhat hastily.
Comments on the Quality of English LanguageThe English language is more comprehensible in this version of the manuscript, the style of writing is still somewhat problematic, but nothing that can't be resolved during editing process.
Author Response
The authors have improved the manuscript: "Association of IFNL4 gene polymorphisms with Hepatitis B Virus (HBV) infection in the Northern Region of Pará, Brazil" to some extent, however, some issues remain. The first is use of healthy control data from a different source, if the authors had no access to the healthy samples, then they should have compared the HBV positive individuals carrying C versus T allele, or compare the 3 genotypes, which brings me to the second issue. Why are two genotypes grouped (TT and TC) while CC is separate? Concluding that T allele can be associated with higher susceptibility to HBV from this data set is somewhat hastily.
We truly appreciate the reviewer’s comments and insights, and agree that, unfortunately, we were unable to analyze a sample of healthy, HBV-uninfected controls using molecular biology techniques.
This limitation led us to compare the genotypes of the infected individuals based on infection criteria, as described in the results. Additionally, we chose to conduct a comparison with a sample group of healthy individuals without infection because this population sample is also from the state of Pará, which reduces the risk of generating non accurate associations that could arise from differences in population substructure patterns, which would likely be the case if a sample from another region of Brazil were used.
Statistical analyses were conducted in order to compare HBV carriers between (i) the IFNL4 gene genotypes, viral infection, and disease severity, with no statistically significant results (“2. Results”, lines 92 to 95: “After this step, we also verified whether there was any possible association between the rs12979860 and the progression/severity of HBV in the individuals comprising our study sample. The results show no statistical significance (p value= 0.9873)”; “Table S2: Paired comparison (Fisher's exact test) of different genotypes and clinical manifestations”); (ii) we compared the presence and absence of liver cirrhosis among these HBV carriers (“2. Results”, lines 115 to 119 “The sample of HBV carriers, when divided into two groups, according to the presence of cirrhosis, were not trustworthy from a statistical analysis standpoint due to the small number of samples from individuals with cirrhosis (N=7). The results of the comparison between the genotypic frequencies and these results are not significant (p value=0.5742)”) Since no statistically significant associations were found, we then proceeded to make comparisons with the healthy population, obtaining statistically interesting results (“2. Results”, lines 105- to 111 “After observing that the rs12979860-T allelic frequency remained high throughout our sample of HBV infection carriers, a comparison between genotypes—the combined genotypes with the T allele (namely, TT and TC) versus the one that is homozygous rs12979860-C—was also carried out, the results of which are statically significant. This confirms that the combination of genotypes with the rs12979860-T allele is indeed prevalent within the group of chronically infected carriers (p value=0.0289).”).
Based on these results, we decided to perform comparisons using the dominant model (TT + TC versus CC) and the recessive model (CC + TC versus TT) to observe whether genotypic combinations could influence the outcomes. The analysis continued to show significant results only for the dominant model, with the low frequency of the mutant genotype (in isolation) in the infected sample, seemingly reflecting this statistically significant difference. The increased frequencies of the reference allele in our sample may suggest a greater risk of infection progression, as described in other studies. We have added these explanations to “2. Results,” lines 88 to 91: “These genotypic frequencies were compared according to the dominant model (p= 0.0037) and the recessive model (p=0.3879) between the samples. The dominant model shows that the homozygous genotype for the rs12979860-C allele had the lowest frequencies in the HBV carrier sample (Table 3),” and in “4. Materials and Methods,” item “4.4. Statistical analysis,” lines 236 to 239: “Comparisons using the dominant model (TT + TC versus CC) and the recessive model (CC + TC versus TT) were also performed within the study sample group, to observe whether genotypic combinations could be influencing the results.”
We also agree with the removal of the statement about the association of the T allele with increased susceptibility to HBV and have modified the text to the following, in lines 145 to 149: “Low genotypic frequencies of the rs12979860-C homozygote allele were observed in the sample of HBV carriers in this study. This result suggests that the high frequencies of the homozygous and heterozygous genotypes for the rs12979860-T allele in the analysis samples may suggest the association of this allele with viral infection, as discussed in previously published studies [14–17].”
Due to these changes, we also modified the text in the Conclusion section, specifically lines 252 and 254: “The sample of HBV carriers from the state of Pará showed high frequencies of the rs12979860-T allele, when compared to the sample of individuals without this viral infection,” and lines 255 to 257: “Future studies should increase the sample size and test new hypotheses about host genetics and factors that may influence the clinical progression of HBV in the state of Pará.” Additionally, we have included three more references that support the discussion section and further exploration of our findings (References section, lines 324 to 335).
- Ren, S.; Lu, J.; Du, X.; Huang, Y.; Ma, L.; Huo, H.; Chen, X.; Wei, L. Genetic Variation in IL28B Is Associated with the Development of Hepatitis B-Related Hepatocellular Carcinoma. Cancer Immunol Immunother 2012, 61, 1433–1439, doi:10.1007/s00262-012-1203-y.
- Chen, J.; Wang, W.; Li, X.; Xu, J. A Meta-Analysis of the Association between IL28B Polymorphisms and Infection Susceptibility of Hepatitis B Virus in Asian Population. BMC Gastroenterol 2015, 15, 2–7, doi:10.1186/s12876-015-0286-2.
- Zhang, Y.; Zhu, S.-L.; Chen, J.; Li, L.-Q. Meta-Analysis of Associations of Interleukin-28B Polymorphisms Rs8099917 and Rs12979860 with Development of Hepatitis Virus-Related Hepatocellular Carcinoma. Onco Tar-gets Ther 2016, 9, 3249–3257, doi:10.2147/OTT.S104904.
- Qin, S.; Wang, J.; Zhou, C.; Xu, Y.; Zhang, Y.; Wang, X.; Wang, S. The Influence of Interleukin 28B Polymorphisms on the Risk of Hepatocellular Carcinoma among Patients with HBV or HCV Infection: An Updated Me-ta-Analysis. Medicine 2019, 98, 1–10, doi:10.1097/MD.0000000000017275.”).
We sincerely hope that after this round of modifications, the manuscript is more adequate for publication.

Reviewer 2 Report
Comments and Suggestions for Authors
The manuscript has been rewritten. Still, some page numbers are missing in the references.
Author Response
The manuscript has been rewritten. Still, some page numbers are missing in the references.
We thank your reviewer feedback, and point out that the required changes and corrections were done as required, with the addition of these pages in the reference section.

Round 3
Reviewer 1 Report
Comments and Suggestions for Authors
The authors have corrected most of my requirements, apart from providing their own healthy controls. If the editors of the journal agree to publish the paper as it is, I will not object anymore.
Comments on the Quality of English LanguageEnglish language still requires moderate editing, but perhaps it is for the best that corrections are done by the journal editors during publishing process.